# Stem: Rethinking Causal Information Flow in Sparse Attention

Lin Niu [* 1]  Xin Luo [* 2 3]  Linchuan Xie [1]  Yifu Sun [1]  Guanghua Yu [1]  Jianchen Zhu [1]  S. Kevin Zhou [4 5 6 7 8]

## Abstract

The quadratic computational complexity of self-attention remains a fundamental bottleneck for scaling Large Language Models (LLMs) to long contexts, particularly during the pre-filling phase. In this paper, we rethink the causal attention mechanism from the perspective of information flow. Due to causal constraints, tokens at initial positions participate in the aggregation of every subsequent token. However, existing sparse methods typically apply a uniform top-$k$ selection across all token positions within a layer, ignoring the cumulative dependency of token information inherent in causal architectures. To address this, we propose **Stem**, a novel, plug-and-play sparsity module aligned with information flow. First, Stem employs the Token Position-Decay strategy, applying position-dependent top-$k$ within each layer to retain initial tokens for recursive dependencies. Second, to preserve information-rich tokens, Stem utilizes the Output-Aware Metric. It prioritizes high-impact tokens based on approximate output magnitude. Extensive evaluations demonstrate that Stem achieves superior accuracy with reduced computation and pre-filling latency.

---

[*]Equal contribution [1]Tencent [2]School of Biomedical Engineering, Division of Life Sciences and Medicine, University of Science and Technology of China, Hefei, Anhui, 230026, China [3]Suzhou Institute for Advanced Research, University of Science and Technology of China, Suzhou, Jiangsu, 215123, China [4]School of Biomedical Engineering, Division of Life Sciences and Medicine, University of Science and Technology of China (USTC), China [5]Medical Imaging, Robotics, Analytic Computing & Learning (MIRACLE) Lab, YRD-RIGHT, USTC Suzhou Institute for Advanced Research, Suzhou, China [6]Jiangsu Provincial Key Laboratory of Multimodal Digital Twin Technology, USTC, China [7]Biomedical Basic Research Center (BBRC) of Jiangsu Province, Suzhou, China [8]State Key Laboratory of Precision and Intelligent Chemistry, USTC, China. Correspondence to: Jianchen Zhu <dickzhu@tencent.com>, S. Kevin Zhou <skevinzhou@ustc.edu.cn>.

*Proceedings of the 43rd International Conference on Machine Learning*, Seoul, South Korea. PMLR 306, 2026. Copyright 2026 by the author(s).

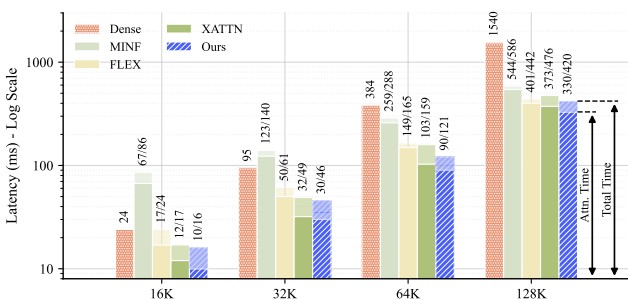

*Figure 1.* Latency comparison (ms) on H20 GPU. Results are reported as Attention Kernel Time/Total Time.

## 1. Introduction

Large Language Models (LLMs) (Vaswani et al., 2017) have revolutionized natural language processing, driving advancements in applications that require extensive context understanding, such as long-document analysis, code generation, and complex agentic workflows (Roumeliotis & Tselikas, 2023; Touvron et al., 2023). However, the quadratic computational complexity of standard self-attention poses a severe challenge for processing increasingly massive input sequences. This bottleneck is particularly acute during the pre-filling phase, where the model must process the entire input prompt in parallel to compute Key-Value (KV) states. As the context window expands, the quadratic scaling of attention causes pre-filling latency and memory overhead to surge disproportionately, creating bottlenecks that impede efficient real-world deployment. To mitigate this quadratic bottleneck, sparse attention mechanisms (Zhang et al., 2023; Xiao et al., 2023; Jiang et al., 2024) have emerged as a promising avenue, aiming to reduce computation by selectively retaining critical Key-Value pairs. While these methods successfully alleviate latency, they often grapple with a significant trade-off between efficiency and model performance. We identify two primary limitations in current paradigms: they prioritize tokens based solely on attention scores and employ a uniform top-$k$ budget across all positions in a layer. We argue that this approach overlooks the causal information flow in the causal architecture: the $n$-th token in a layer is built by aggregating from the first to the $n$-th token of the preceding layer. Consequently, tokens at initial positions participate in the aggregation of every subsequent token, effectively accumulating information recursively across layers. Indiscriminately pruning tokens

at the initial position disrupts signal propagation to deeper layers. Moreover, existing methods select tokens based on simulated attention scores, rather than the actual information contribution. These factors collectively degrade the accuracy of sparsified models.

To address these limitations, we propose **Stem**, a novel training-free framework that treats early tokens as the structural "stem" of information flow to optimize the pre-filling phase. Our approach is motivated by a theoretical analysis of how sparsification affects the model's final output, revealing that the integrity of the recursive dependency chain is paramount. Guided by this insight, we introduce the Token Position-Decay (TPD) strategy. Unlike uniform selection, TPD dynamically adjusts the sparse budget—defined as the ratio of computed token pairs to full attention; Specifically, it allocates a larger budget to tokens at the initial positions in a layer, while aggressively sparsifying the tokens at later positions. Furthermore, to minimize the loss between the sparse and dense attention output, we propose the Output-Aware Metric (OAM). OAM no longer relies solely on attention scores, but simulates the output magnitude (incorporating value information), ensuring that tokens containing important value information are retained. Thus, Stem delivers superior accuracy using a reduced computational budget, implemented via the open-source Block Sparse Attention kernel (Guo et al., 2024) for efficient execution.

We evaluate our method on benchmarks like RULER (Hsieh et al., 2024) and LongBench (Bai et al., 2024) using Llama-3.1-8B (Dubey et al., 2024) and Qwen3-8B (Yang et al., 2025). Results show that Stem consistently outperforms existing training-free sparse attention methods in both accuracy and latency. Moreover, our framework functions as a plug-in that can be integrated into training-based sparse models, such as DeepSeek-V3.2 (Liu et al., 2025) and MiniCPM-4.1 (Team et al., 2025), to further compress the sparse budget. Extensive experiments demonstrate that Stem improves sparsity rates and accelerates Time-to-First-Token (TTFT) while maintaining lossless precision. Figure 1 highlights the superior latency performance of Stem.

Our contributions are summarized as follows:

- We rethink sparse attention from the perspective of causal information flow, identifying the inter-layer recursive dependency as a critical factor often neglected by static selection methods.

- We propose Stem, a training-free framework comprising a Token Position-Decay strategy to preserve the causal dependency chain and an Output-Aware Metric to simulate output magnitude.

- Empirical results verify that Stem outperforms training-free methods and enhances training-based models,

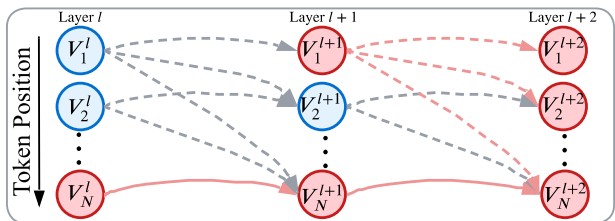

*Figure 2.* **Visualization of recursive error propagation.** The diagram depicts the impact of sparsification across layers $l \rightarrow l+2$ based on Eq. (1). Red circles indicate sparse tokens, while blue circles indicate dense tokens. Pruning the initial token $V_1^{(l+1)}$ triggers a global distortion that affects all tokens in the next layer (dashed red connections), whereas pruning the last token $V_N^{(l)}$ results in only a local error confined to the tail.

achieving higher efficiency with our open-source Triton implementation.

## 2. Methodology

This section introduces Stem, starting with the Token Position-Decay strategy (Section 2.1) derived from recursive flow analysis. This is followed by the Output-Aware Metric (Section 2.2) for optimized token selection.

### 2.1. Token Position-Decay

**Theoretical Analysis of Information Flow.** Consider the $l$-th Transformer layer, where $l \in \{1, \ldots, L\}$ and $L$ denotes the total number of layers, with input embeddings $X \in \mathbb{R}^{N \times d}$ with sequence length $N$ and hidden dimension $d$. Let $P^{(l)} \in \mathbb{R}^{N \times N}$ be the causal attention probability matrix and $V^{(l)} \in \mathbb{R}^{N \times d}$ be the Value, so the attention output is given by $O^{(l)} = P^{(l)}V^{(l)}$ with $O^{(l)} \in \mathbb{R}^{N \times d}$. Under the causal constraint, for a query position $i$ and a source position $j$ satisfying $1 \leq j \leq i \leq N$, the output at position $i$ is strictly a weighted sum of preceding Value vectors, formulated as $O_i^{(l)} = \sum_{j=1}^{i} P_{i,j}^{(l)} V_j^{(l)}$. This formulation reveals a fundamental asymmetry in token participation:

- The first Value vector, $V_1^{(l)}$, appears as a constituent term in the computation of every output $\{O_i^{(l)}\}_{i=1}^{N}$.

- In contrast, the last Value vector, $V_N^{(l)}$, participates only in the computation of the final output $O_N^{(l)}$.

This intra-layer asymmetry is further amplified across layers. In a deep Transformer, the Value vectors for the next layer, $V^{(l+1)}$, are derived from the current layer's outputs via a composite mapping $\mathcal{T}$ (encompassing FFN, residuals, and the Value projection $W_V$). To formulate this recursive dependency, we express the transition for layer $l+1$ as:

$$V_i^{(l+1)} = \mathcal{T}\left(O_i^{(l)}\right) = \mathcal{T}\left(\sum_{j=1}^{i} P_{i,j}^{(l)} V_j^{(l)}\right)$$

$$\implies \quad V_1^{(l+1)} = \mathcal{T}\left(P_{1,1}^{(l)} V_1^{(l)}\right)$$

$$V_2^{(l+1)} = \mathcal{T}\left(P_{2,1}^{(l)} V_1^{(l)} + P_{2,2}^{(l)} V_2^{(l)}\right) \qquad (1)$$

$$\vdots$$

$$V_N^{(l+1)} = \mathcal{T}\left(P_{N,1}^{(l)} V_1^{(l)} + \cdots + P_{N,N}^{(l)} V_N^{(l)}\right)$$

The expansion in Eq. (1) demonstrates the asymmetric sensitivity of the network state to token pruning. Let us consider the consequence of sparsifying (i.e., removing) a specific token $V_j^{(l)}$ from the attention computation:

- **Sparse at the Initial Position** ($j = 1$): If $V_1^{(l)}$ is discarded, the term $P_{i,1} V_1^{(l)}$ vanishes from the summation in every row of the system in Eq. (1). As illustrated by the dashed red connections in Figure 2, this results in a global distortion that corrupts the representation of all tokens $V_1^{(l+1)}, \ldots, V_N^{(l+1)}$ in the next layer. This error accumulates and amplifies recursively across layers. So, sparsifying early tokens causes substantial downstream errors.

- **Sparse at the Last Position** ($j = N$): In contrast, if $V_N^{(l)}$ is discarded, the error is confined strictly to the computation of the final vector $V_N^{(l+1)}$ (indicated by the dense red path in Figure 2). The representations of all preceding tokens $V_{1:N-1}^{(l+1)}$ remain entirely unaffected. So, sparsifying later tokens affects few outputs.

This formulation identifies a recursive accumulation effect. Since $V_1^{(l)}$ is embedded in every output $O^{(l)}$, and these outputs serve as the source for the next layer's inputs, the information encoded in initial position tokens propagates to the entire sequence representation in deeper layers. From an information flow perspective, initial position tokens act as recursive anchors. If $V_1$ is sparsified at layer $l$, the error propagates to all $N$ tokens in layer $l + 1$, compounding recursively. Thus, maintaining high fidelity for initial position tokens is paramount for global model accuracy.

To validate this, we analyze the error distribution using Qwen3-8B on 30 8K-length cases from LongBench (Figure 3). Results show that applying sparse attention to the initial position ($[0, 2k)$) incurs significantly higher head logits loss compared to the last position ($[6k, 8k)$). This initial sensitivity persists across both fixed and dynamic sparsity configurations. This confirms our theory that initial position act as recursive anchors, where information loss amplifies

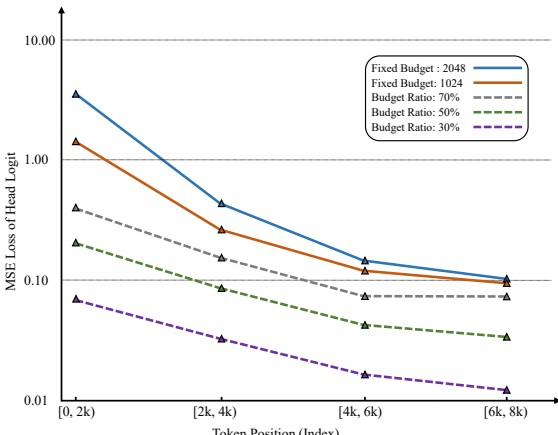

*Figure 3.* **Sensitivity analysis of token position segments.** The X-axis represents the specific token interval subject to sparsification. Y-axis shows head logit MSE loss. Curves compare dynamic ratios vs. fixed budgets.

globally. Conversely, the high tolerance of later positions motivates our strategy to allocate higher budgets to initial position tokens.

**Dynamic Sparse Budget.** As illustrated in Figure 4 (a), existing methods typically employ a uniform Top-$k$ budget ($k_{\text{uni}}$), imposing a constant allocation upper bound. However, as shown by the light-blue shaded "causal triangle", the actual cost is lower at the start. For a sequence of length $N$, the resulting total computational cost is:

$$\mathcal{C}_{\text{uni}} \approx N \cdot k_{\text{uni}} - \frac{1}{2} k_{\text{uni}}^2 \qquad (2)$$

where the term $-\frac{1}{2} k_{\text{uni}}^2$ accounts for the "causal triangle" at the beginning of the sequence, i.e., positions with $i < k_{\text{uni}}$ have fewer than $k_{\text{uni}}$ available past tokens under causal masking.

To align sparsity with information flow, our strategy linearly decays the budget from an initial $k_{\text{start}}$ to $k_{\text{end}} = \mu \cdot k_{\text{start}}$ with $\mu \in (0, 1]$, as shown in Figure 4 (b). The per-position budget then decays linearly from $k_{\text{start}}$ to $k_{\text{end}}$ over the sequence. Concretely, for the query at position $i \in \{1, \ldots, N\}$, we set the Top-$k$ budget via linear interpolation as:

$$k(i) = \left\lfloor k_{\text{start}} - \left(\frac{k_{\text{start}}(1 - \mu)}{N}\right) \cdot i \right\rfloor. \qquad (3)$$

Based on this schedule, the total computational cost of our decay strategy $\mathcal{C}_{\text{decay}}$ can be derived as:

$$\mathcal{C}_{\text{decay}} \approx \underbrace{N k_{\text{start}} - \frac{1}{2} k_{\text{start}}^2}_{\text{Uniform Baseline}} - \underbrace{\frac{1}{2} k_{\text{start}}(1 - \mu)(N - k_{\text{start}})}_{\text{Decay Savings}}.$$
$$(4)$$

This strategy ensures that critical early positions are retained with a large budget ($k_{\text{start}}$), preserving recursive integrity,

while later positions are progressively pruned according to $\mu$. The final term explicitly quantifies the computational savings relative to the uniform baseline, i.e., $\mathcal{C}_{\text{uni}}$ with $k_{\text{uni}} = k_{\text{start}}$. This reduction reflects the aggressive pruning of later redundancy, improving the trade-off between information fidelity and inference efficiency. Empirically, we adopt $\mu = 0.7$ to balance these factors, with detailed sensitivity analysis in ablation studies (Figure 5).

## 2.2. Output-Aware Metric

While the Token Position-Decay strategy allocates the sparse budget across positions, the criterion for selecting specific tokens remains critical. Let $Q, K, V \in \mathbb{R}^{N \times d}$ denote the Query, Key, and Value matrices. Existing training-free methods rely on a Score-Aware Metric (SAM) that approximates the scaled routing score $Q_i K_j^\top / \sqrt{d}$ as a proxy. For instance, MInference utilizes recent queries for approximation, while XAttention and MiniCPM-4 estimate scores via downsampled block-wise representations. We follow this paradigm by adopting block-wise downsampling for metric calculation, as it aligns seamlessly with FlashAttention kernels for efficient hardware execution.

**Limitations of Score-Aware Metric.** We argue that attention scores capture only the *routing probability*, not the actual *information contribution*. Recall that the attention mechanism computes a weighted sum of Value vectors: $O_i = \sum_j P_{i,j} V_j$. A token $j$ might have a high score $P_{i,j}$ with the query, yet if its Value vector $V_j$ has a negligible magnitude (i.e., $\|V_j\| \approx 0$), its actual contribution to the output $O_i$ is minimal. Conversely, a token with a moderate score but a high-magnitude Value vector ("high-energy signal") can significantly influence the residual stream. Pruning such tokens based solely on scores introduces large approximation errors.

**Metric Derivation.** Our objective is to select a subset of indices $\mathcal{S}$ such that the reconstruction error between the sparse output $\tilde{O}_i$ and the dense output $O_i$ is minimized:

$$\min_{\mathcal{S}} \|O_i - \tilde{O}_i\|_2 = \min_{\mathcal{S}} \left\| \sum_{j \notin \mathcal{S}} P_{i,j} V_j \right\|_2 \quad (5)$$

To minimize this error, the optimal strategy is to retain the tokens that maximize the magnitude of the individual contribution term $\|P_{i,j} V_j\|_2$. Since the attention probability scales with the exponential of the dot product (i.e., $P_{i,j} \propto \exp(Q_i K_j^T / \sqrt{d})$), the magnitude of a token's contribution is proportional to $\exp(Q_i K_j^T / \sqrt{d}) \cdot \|V_j\|_2$.

Directly computing this product involves expensive exponential operations. To construct a computationally efficient metric that is compatible with standard Top-$k$ kernels, we

apply a logarithmic transformation. Since the logarithm is monotonic, it preserves the ranking order:

$$\mathcal{M}_{i,j} = \log \left( \exp \left( \frac{Q_i K_j^T}{\sqrt{d}} \right) \cdot \|V_j\|_2 \right)$$
$$= \frac{Q_i K_j^T}{\sqrt{d}} + \log(\|V_j\|_2) \quad (6)$$

We refer readers to Appendix A.1 for the full derivation.

**The Proposed Metric.** Based on this derivation, we define the **Output-Aware Metric (OAM)** for token selection as:

$$\mathcal{M}_{i,j} = \underbrace{Q_i K_j^T}_{\text{Routing}} + \beta \cdot \underbrace{\max(0, \log(\|V_j\|_2))}_{\text{Magnitude}} \quad (7)$$

This formulation combines two critical factors into a unified score: **Routing Relevance** ($QK^T$), which aligns with the standard Score-Aware Metric (SAM), and **Signal Magnitude**, which captures the information density of the value vector. We introduce a non-negative constraint $(\max(0, \cdot))$ and a coefficient $\beta$ to regulate scales, ensuring magnitude acts as a differentiator without overshadowing routing semantics. Empirically, we set $\beta = 0.2$ based on sensitivity analysis (Figure 5).

To validate OAM's superiority over SAM, we evaluate Qwen3-8B on 30 LongBench samples (8K length) with a fixed top-$k$ budget of 1024 tokens. We measure the sparse-dense MSE loss at varying depths (e.g., **L5** denotes layer 5) alongside the final head logit loss. As shown in Table 1, OAM consistently achieves lower reconstruction errors. This confirms that incorporating $\log \|V\|_2$ successfully retains high-impact tokens, even those with moderate routing scores, effectively mitigating information loss.

*Table 1.* **Comparison of sparse loss between SAM and OAM.**

| Method | L5 | L15 | L25 | L35 | Head Logits |
|--------|------|------|------|------|-------------|
| SAM | 1.5e-5 | 2.0e-4 | 8.5e-4 | 6.0e-3 | 0.3368 |
| **OAM** | **1.4e-5** | **2.0e-4** | **8.1e-4** | **5.2e-3** | **0.3126** |

## 2.3. Overall Algorithm

We integrate the proposed Position-Decay strategy and Output-Aware Metric into a unified, training-free framework: Stem. The detailed inference procedure is outlined in Algorithm 1.

As illustrated in Figure 4, the algorithm operates in a coarse-to-fine manner across three stages, utilizing the Block Sparse Attention library (Guo et al., 2024) for efficient sparse kernel execution. **First**, to align with kernel granularity and maximize hardware utilization, we fix the block size $B = 128$ and, following XAttention (Xu et al., 2025),

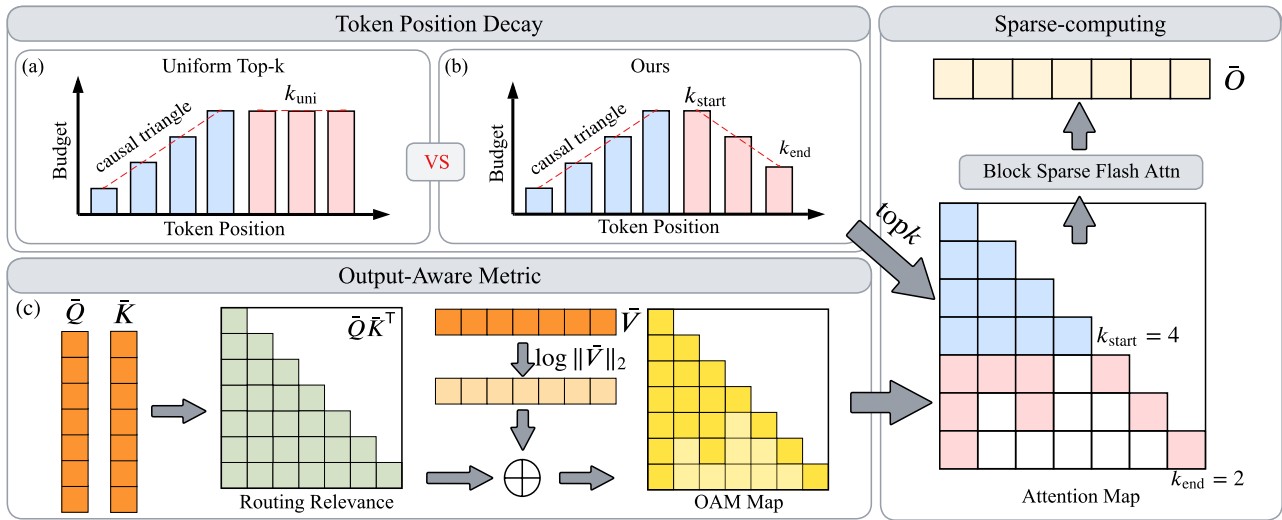

*Figure 4.* **Pipeline of Stem.** The figure illustrates the framework workflow and compares the sparsity budget schedules between the standard Uniform Top-$k$ and our Token Position-Decay strategy.

adopt anti-diagonal scoring for efficient metric downsampling of Query and Key matrices ($\bar{Q}, \bar{K}$), alongside max-pooled Value magnitudes ($\bar{M}_V$). This downsampling reduces metric computation overhead by a quadratic factor of $B^2$. **Second**, we derive the dynamic budget schedule ($k_{start}, \mu$) to allocate higher budgets to critical initial dependencies. **Third**, during the inference loop, we compute the Output-Aware Metric $\mathcal{M}$ against $\bar{K}$ to identify the most significant memory blocks. Only the full-resolution blocks corresponding to the selected Top-$k(i)$ indices are retrieved for exact Softmax and aggregation. This strategy ensures linear complexity while preserving high-fidelity information flow.

In summary, Stem transforms the quadratic complexity bottleneck into a manageable linear scale dictated by the hyperparameters $k_{start}$ and $\mu$, while the OAM ensures that this compression does not sacrifice the information flow essential for model accuracy.

## 3. Experiment

This section comprehensively evaluates Stem's accuracy and prefill efficiency, detailing experimental settings, main results, ablation studies, and visualizations.

### 3.1. Experimental Settings

**Models and Datasets.** We select Llama-3.1-8B-Instruct and Qwen3-8B as dense backbones to evaluate the effectiveness of our training-free module. We also utilize DeepSeek-V3.2-671B and MiniCPM-4.1-8B to evaluate budget compression when integrated into models with trained sparsity.

We conduct experiments on LongBench for practical bilingual evaluation and RULER for controllable synthetic stress testing. Together, they cover both synthetic and realistic task distributions. All evaluations utilize the open-source `lm-evaluation-harness` framework (Gao et al., 2021).

**Implementation Details.** All experiments are conducted on NVIDIA H20 GPUs using PyTorch. We leverage FlashAttention2-CUDA for dense models and the Block Sparse Attention library for our sparse approach. Regarding sparsity hyperparameters, we set block-wise initial budget $k_{start}$ to $0.2N_{blk}$ for sequence lengths of 8k–16k and $0.1N_{blk}$ for lengths exceeding 16k, where $N_{blk} = \lceil N/B \rceil$ denotes the total number of blocks. To ensure stability, we allocate 4 blocks for both initial and local windows, enforcing a minimum total budget of 54 blocks. Finally, we employ an aggressive decay ratio of $\mu = 0.7$ and set the balancing coefficient $\beta = 0.2$ in the Output-Aware Metric to regulate the numerical scales of Routing and Magnitude terms. We define the reported sparsity budget as the ratio of retained attention blocks to all admissible attention blocks under the causal mask. This block-retention ratio directly reflects the sparse attention computation executed by the kernel, while end-to-end latency is reported separately to account for metric computation, index generation, block retrieval, and framework overhead.

**Baselines.** We use the standard dense attention as a reference and compare against representative training-free and training-based sparse methods. MInference (MINF) and FlexPrefill (FLEX) are training-free approaches that rely on sparse pattern selection and calibration-based budget allocation to reduce computation. XAttention (XATTN) performs block-level pruning with hyperparameter-controlled adaptive sparsity, enabling plug-and-play acceleration. DSA

**Algorithm 1 Stem**

1: **Input:** Query $Q$, Key $K$, Value $V \in \mathbb{R}^{N \times d}$
2: **Hyperparameters:** Initial Budget $k_{\text{start}}$, Decay Ratio $\mu$, Block Size $B$, Coeff $\beta$
3: **Output:** Attention Output $O \in \mathbb{R}^{N \times d}$
4: # 1. Pre-compute Block-wise Representations
5: $\bar{Q}, \bar{K} \leftarrow \text{Pool}(Q, K, \text{stride} = B)$    # Anti-diagonal scoring downsampling
6: $\bar{M}_V \leftarrow \text{MaxPool}(\log(\|V\|_2), \text{stride} = B)$    # Max-pooling for Value Magnitude
7: # 2. Determine Position-Decay Schedule
8: $k_{\text{end}} \leftarrow \mu \cdot k_{\text{start}}$
9: # 3. Block-wise Coarse-to-Fine Inference
10: **for** each query block index $i$ from 1 to $\lceil N/B \rceil$ **do**
11:    # a. Estimate Coarse Metric via Downsampled Blocks (Matches Eq. 7)
12:    $\bar{S}_{\text{route}} \leftarrow \bar{Q}_i \bar{K}^T$    # Routing term: $Q_i K_j^T$
13:    $\bar{\mathcal{M}}_i \leftarrow \bar{S}_{\text{route}} + \beta \cdot \max(0, \bar{M}_V^T)$    # Metric = Routing + $\beta \cdot$ Magnitude
14:    # b. Determine Dynamic Budget (in number of blocks)
15:    $k_{curr} \leftarrow \lfloor \text{Interp}(k_{\text{start}}, k_{\text{end}}, i)/B \rfloor$
16:    # c. Select Top-k Blocks based on Metric $\bar{\mathcal{M}}$
17:    $\mathcal{I}_{\text{block}} \leftarrow \text{TopK}(\bar{\mathcal{M}}_i, k = k_{curr})$
18:    # d. Fine-grained Sparse Aggregation (Exact Computation)
19:    $K_{\text{sparse}}, V_{\text{sparse}} \leftarrow \text{Gather}(K, V, \text{indices} = \mathcal{I}_{\text{block}})$
20:    $S_{\text{exact}} \leftarrow Q_i K_{\text{sparse}}^T / \sqrt{d}$    # $Q_i$ is full-resolution block
21:    $P_{\text{sparse}} \leftarrow \text{Softmax}(S_{\text{exact}})$
22:    $O_i \leftarrow P_{\text{sparse}} V_{\text{sparse}}$
23: **end for**
24: **Return** $O$

represents trained structured sparse attention, performing token-level sparse computation by selecting the top-2048 elements from the attention map. InfLLM v2 utilizes a block-wise sparse mechanism, selecting a fixed budget of 96 blocks to accelerate inference.

### 3.2. Main Results

**LongBench.** Table 2 presents the LongBench results. While MInference maintains accuracy, it demands excessive budgets (69%–81%). Conversely, aggressive methods like FlexPrefill and XAttention suffer significant degradation, particularly on multi-document (MD) and synthetic (SYN) tasks, highlighting the limitations of current pruning heuristics. In contrast, Stem achieves the highest accuracy among sparse baselines despite using the lowest budget (25%–31%). Notably, it outperforms the runner-up on Qwen3-8B by over 1% and nearly matches dense performance on Llama-3.1-8B-Instruct (41.48% vs. 42.02%). This validates our design

*Table 2.* **LongBench results (%).** We compare accuracy on Llama-3.1-8B-Instruct and Qwen3-8B across task families: Code Completion (CC), Few-Shot Learning (FSL), Multi-Document QA (MD1/MD2), Summarization (SUM), and Synthetic (SYN). AVG and BUD denotes the average accuracy and the sparsity budget.

| METHOD | CC | FSL | MD1 | MD2 | SUM | SYN | AVG | BUD. |
|---|---|---|---|---|---|---|---|---|
| **QWEN3-8B** | | | | | | | | |
| DENSE | 19.09 | 63.10 | 11.23 | 15.06 | 20.63 | 62.92 | 32.01 | 100% |
| MINF | 18.91 | 61.87 | 10.78 | 14.46 | 20.26 | 55.32 | 30.27 | 69% |
| FLEX | 19.75 | 55.31 | 10.76 | 13.49 | **21.18** | 50.83 | 28.55 | 31% |
| XATTN | **21.03** | 62.30 | **11.42** | 14.64 | 20.47 | 52.92 | 30.46 | 28% |
| STEM | 19.43 | 61.84 | 11.22 | **14.97** | 20.21 | **62.19** | **31.64** | **25%** |
| **LLAMA-3.1-8B-INSTRUCT** | | | | | | | | |
| DENSE | 36.08 | 64.29 | 27.54 | 31.19 | 25.10 | 67.92 | 42.02 | 100% |
| MINF | 35.43 | 62.98 | 25.42 | 29.00 | 25.12 | **68.41** | 41.06 | 81% |
| FLEX | 34.65 | 59.51 | 17.17 | 21.62 | 24.46 | 59.10 | 36.09 | 34% |
| XATTN | **37.23** | **63.61** | 22.15 | 28.23 | **25.30** | 50.95 | 37.91 | 35% |
| STEM | 35.86 | 62.89 | **26.33** | **30.53** | 24.93 | 68.32 | **41.48** | **31%** |

rationale: preserving initial tokens safeguards long-range dependency chains, while value-aware selection captures signal-rich tokens. Thus, Stem provides robust sparsification, delivering superior accuracy even under the most aggressive reduction.

To demonstrate versatility and orthogonality, we integrate Stem into two training-based sparse models: DeepSeek-V3.2 (DSA) and MiniCPM-4 (InfLLMv2). These experiments are not intended to improve the accuracy of native sparse models; instead, they test whether Stem can further reduce the computation budget with negligible performance change. These models natively learn to select top-$k$ tokens or blocks during training. For DeepSeek-V3.2, the baseline DSA selects the top-2048 tokens based on QK scores. By applying Stem (using the Output-Aware Metric and Token Position-Decay with $k_{\text{start}} = 2048, \mu = 0.7$) on top of the native DSA kernel, we achieved a 15% reduction in the average sparsity budget. Similarly, for MiniCPM-4, which natively maintains initial and local blocks while selecting the top-64 blocks, we replaced the fixed top-$k$ selection with **Stem**'s decay schedule ($k_{\text{start}} = 64$ blocks, $\mu = 0.7$). This integration resulted in an 18% reduction in the computational budget. As presented in Table 3, Stem maintains accuracy comparable to the original sparse baselines across all LongBench categories despite the reduced budget. This indicates that even in models explicitly trained for sparsity, our information-flow-driven approach can identify and prune residual redundancy without compromising performance.

**RULER.** Table 4 reports RULER results across context lengths (4K–128K). Stem distinguishes itself by achieving the highest average accuracy across both models while maintaining the strictly lowest sparsity budget (25%). Notably, while MINF attains high accuracy in specific settings, it re-

*Table 3.* **Performance on training-based sparse models.**

| METHOD | CC | FSL | MD1 | MD2 | SUM | SYN | AVG |
|---|---|---|---|---|---|---|---|
| **DEEPSEEK-V3.2** | | | | | | | |
| DSA | **32.42** | 41.85 | **51.83** | 48.30 | **22.28** | 60.35 | 42.84 |
| + STEM | 32.02 | **44.03** | 51.67 | **48.44** | 21.97 | **60.85** | **43.16** |
| **MINICPM-4.1** | | | | | | | |
| INFLLMV2 | 55.61 | **53.74** | 42.83 | **20.26** | 17.10 | 67.00 | 42.75 |
| + STEM | **55.62** | 53.65 | **43.03** | 20.13 | **17.13** | 67.00 | **42.76** |

*Table 4.* **Accuracy (%) and Budget on the RULER benchmark.**

| METHOD | 4K | 8K | 16K | 32K | 64K | 128K | AVG | BUD. |
|---|---|---|---|---|---|---|---|---|
| **LLAMA-3.1-8B-INSTRUCT** | | | | | | | | |
| DENSE | 95.52 | 94.06 | 93.43 | 88.19 | 84.86 | 77.11 | 88.86 | 100% |
| MINF | **95.51** | 93.56 | 92.70 | **91.68** | 80.06 | **76.67** | 88.36 | 55% |
| FLEX | 94.98 | 93.91 | 92.68 | 89.51 | 82.66 | 75.40 | 88.19 | 27% |
| XATTN | 95.46 | 93.83 | 93.65 | 89.82 | 81.60 | 74.35 | 88.12 | 26% |
| STEM | 95.17 | **93.96** | **93.73** | 89.92 | **82.67** | 75.37 | **88.47** | **25%** |
| **QWEN3-8B** | | | | | | | | |
| DENSE | 95.44 | 93.08 | 92.71 | 92.12 | 79.57 | 73.03 | 87.66 | 100% |
| MINF | 95.01 | 92.56 | 91.70 | 91.68 | 77.54 | 71.87 | 86.73 | 76% |
| FLEX | 95.06 | 91.98 | 91.17 | 90.93 | 77.31 | 70.28 | 86.12 | 30% |
| XATTN | 95.03 | 92.56 | 91.76 | 91.72 | 78.60 | 71.80 | 86.91 | 35% |
| STEM | **95.18** | **92.68** | **92.11** | **91.85** | **78.78** | **72.31** | **87.15** | **25%** |

quires a much larger budget (55%–76%), consuming $2\times$ to $3\times$ more memory than our approach. Stem overcomes this trade-off, delivering near-lossless performance compared to the dense baseline with minimal computational overhead. This efficiency is driven by the synergy of our position-decay strategy, which safeguards recursive dependencies, and the output-aware metric, which prioritizes tokens with the highest semantic value.

### 3.3. Ablation Studies

To isolate the contributions of the Token Position Decay (TPD) strategy and the Output-Aware Metric (OAM), we conducted an ablation study on LongBench using both Llama-3.1-8B-Instruct and Qwen3-8B. Crucially, to ensure a fair comparison, we maintained a strictly consistent total sparsity budget across all settings. Specifically, for the Uniform baseline, we derive a constant top-$k$ budget, denoted as $k_{uni}$, such that its total computational cost matches that of the TPD strategy parameterized by $k_{start}$ and $\mu$. This relationship is formulated as $k_{uni} \approx (k_{start} + k_{end})/2 = k_{start} \cdot (1+\mu)/2$. Given $\mu = 0.7$, it follows that $k_{uni} = 0.85 \cdot k_{start}$. This implies that the Uniform baseline distributes the budget evenly, whereas TPD reallocates resources to prioritize initial positions.

As shown in Table 5, the Uniform baseline (using standard SAM) yields the lowest performance (e.g., 29.41% on Qwen3), indicating that a uniform distribution fails to preserve critical recursive dependencies at the start. Introducing +TPD significantly boosts accuracy (e.g., +2.02% on

*Table 5.* **Ablation study on LongBench.**

| CONFIG | CC | FSL | MD1 | MD2 | SUM | SYN | AVG |
|---|---|---|---|---|---|---|---|
| **QWEN3-8B** | | | | | | | |
| UNIFORM | 19.01 | 58.92 | 10.91 | 14.32 | 20.12 | 53.23 | 29.41 |
| + TPD | 19.28 | 60.81 | **11.42** | **15.04** | 20.46 | **61.58** | 31.43 |
| + TPD + OAM | **19.43** | **61.03** | 11.34 | 14.94 | 20.44 | 60.92 | **31.64** |
| **LLAMA-3.1-8B-INSTRUCT** | | | | | | | |
| UNIFORM | 34.02 | 57.26 | 21.96 | 28.21 | 23.61 | 59.86 | 37.48 |
| + TPD | 35.62 | 61.70 | **26.50** | 28.83 | 24.01 | **68.43** | 40.85 |
| + TPD + OAM | **35.86** | **62.89** | 26.33 | **30.53** | **24.93** | 68.32 | **41.48** |

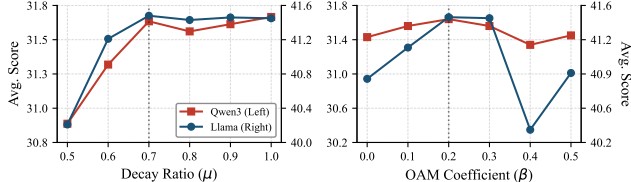

*Figure 5.* Hyperparameter ablation studies on LongBench.

Qwen3, +3.37% on Llama-3.1) under the exact same budget. This confirms that structurally preserving early tokens is more effective than a uniform allocation. Furthermore, incorporating +OAM (our full Stem framework) delivers additional gains (reaching 31.64% and 41.48%), validating that magnitude-aware selection captures information flow more faithfully than routing scores alone. The different gain patterns of TPD and OAM are expected because they address different failure modes. TPD corrects a structural bias of uniform pruning: under causal attention, errors from early positions propagate to many later representations, so reallocating budget toward early positions yields the dominant improvement. OAM, in contrast, refines token ranking after the budget has been set by incorporating value magnitude. Its gain is therefore more fine-grained and task-dependent: it is most helpful when useful evidence is distributed across several candidate tokens, while on precise localization tasks the original QK ranking can already be close to optimal and the magnitude term may bring smaller gains.

**Impact of Decay Ratio** ($\mu$)**.** We varied $\mu$ from 0.5 to 1.0 (where 1.0 represents a uniform budget). As shown in Figure 5 (Left), model performance improves initially as $\mu$ increases but saturates around $\mu = 0.7$. Notably, the accuracy at $\mu = 0.7$ is nearly identical to the uniform baseline ($\mu = 1.0$) for both Qwen3 and Llama-3.1 (e.g., 31.64% vs. 31.67% on Qwen3), yet it significantly reduces the computational cost. However, dropping $\mu$ below 0.6 leads to a sharper performance decline, suggesting that overly aggressive pruning of tail tokens begins to damage local context integrity. Thus, $\mu = 0.7$ serves as the optimal Pareto point between efficiency and accuracy.

**Impact of Balancing Coefficient** ($\beta$)**.** We analyzed the weight $\beta$ in the metric $\mathcal{M} = S_{route} + \beta \cdot \max(0, \log(\|V_j\|_2))$, ranging from 0 to 0.5. Figure 5 (Right) reveals a clear uni-

modal trend. Starting from $\beta = 0$ (pure SAM), performance consistently improves as we introduce magnitude awareness, peaking at $\beta = 0.2$. This confirms that value magnitude provides a useful signal for token importance. However, as $\beta$ exceeds 0.3, accuracy begins to degrade. This indicates that an excessive weight on magnitude may introduce noise, overshadowing the semantic alignment captured by the routing scores. Consequently, we select $\beta = 0.2$ as the robust default.

**Theoretical Complexity.** Let $N$, $d$, and $B$ denote the sequence length, hidden dimension, and block size. Standard dense attention incurs a complexity of $\mathcal{O}(4N^2d + 3N^2)$ (accounting for matrix multiplications and element-wise softmax operations). In contrast, Stem decomposes the computation into *Metric Calculation* and *Sparse Execution*. Given the average budget $k_{\text{avg}} = \frac{1}{N} \sum_{i=1}^{N} k(i)$, we formulate the total complexity $\mathcal{C}_{\text{Stem}}$ as:

$$\mathcal{C}_{\text{Stem}} \approx \underbrace{\frac{2N^2d}{B^2} + \frac{Nd}{B}}_{\text{Metric Calculation}} + \underbrace{4Nk_{\text{avg}}d + 3Nk_{\text{avg}}}_{\text{Sparse Attn. Cal.}} \quad (8)$$

The Metric Calculation term reduces the quadratic routing overhead by a factor of $B^2$ via downsampling, while the magnitude term ($\log \|V\|$) is computed at the block level, incurring only linear $Nd/B$ overhead. Consequently, the additional overhead introduced by Metric Calculation is negligible. The Sparse Attention term scales linearly with $N$ since the average budget $k_{\text{avg}} \ll N$. Consequently, Stem theoretically achieves linear scaling while mitigating the memory I/O bottleneck.

**Empirical Latency.** We use the standard FlashAttention-2 (CUDA version) as the dense baseline on Llama-3.1-8B-Instruct. Unless otherwise noted, all latency measurements are run with batch size 1, BF16, and native PyTorch. Figure 1 reports the Kernel / Total Latency Execution Time across varying context lengths (16K–128K). At 128K context, Stem reduces latency from 1540ms (Dense) to 420ms, achieving a $3.7\times$ speedup. The metric calculation overhead for Stem at 128K is approx. 90ms, which is significantly slower than MInference (approx. 40ms overhead but much slower execution) and comparable to XAttention, validating the efficiency of our block-wise downsampling. MInference exhibits high latency at shorter contexts (e.g., at 16K/32K, it is slower than Dense) due to costly pattern estimation. FlexPrefill and XAttention perform well, but Stem consistently achieves the lowest latency across all lengths. This is attributed to our Position-Decay strategy, which effectively lowers the average budget $k_{\text{avg}}$ without compromising accuracy, allowing for faster sparse execution.

## 4. Related Work

### 4.1. Efficient Long-Context LLMs

The quadratic complexity of self-attention is a major barrier to processing long contexts in LLMs. To mitigate this, system and memory optimizations have been developed. These include I/O-aware kernels like FlashAttention (Dao et al., 2022), multi-device strategies like RingAttention (Liu et al., 2023), and memory management techniques like PagedAttention (Kwon et al., 2023) and Multi-Head Latent Attention (MLA) (Liu et al., 2024). While effective for improving speed and memory usage, these approaches do not alter the fundamental $O(N^2)$ algorithmic complexity of attention.

### 4.2. Linear Attention

To bypass the quadratic bottleneck, researchers have explored architectures with linear complexity. More recently, State Space Models (SSMs) like Mamba (Gu & Dao, 2024), and linear Recurrent Neural Networks (RNNs) such as RWKV (Peng et al., 2023), RetNet (Roy et al., 2023), Gated DeltaNet (Yang et al., 2024), and Log-Linear Attention (Guo et al., 2025), have gained popularity for their efficiency in handling long sequences. Despite their efficiency, these models often trail full-attention Transformers in in-context learning and precise retrieval capabilities, particularly on tasks requiring information extraction from distant tokens. Consequently, optimizing standard attention via sparsity remains indispensable for high-performance modeling.

### 4.3. Sparse Attention

Sparse attention approximates dense attention by computing only significant token interactions, broadly categorized into training-free and training-based methods.

**Training-free Methods.** These methods prune attention connections at inference time without modifying model weights. Early works like H2O (Zhang et al., 2023) and TOVA (Oren et al., 2024) prune the KV cache by discarding tokens based on cumulative query patterns, while StreamingLLM (Xiao et al., 2023) employs a static strategy that retains initial and recent tokens to maintain stability across extended sequences. Recent advancements focus on dynamic, input-dependent sparsity. For instance, MInference (Jiang et al., 2024) and FlexPrefill (Lai et al., 2025) accelerate the prefill stage by dynamically selecting patterns (e.g., Vertical-Slash or block sparsity) and simulating score distributions to filter top-$k$ tokens for the attention kernel; meanwhile, XAttention (Xu et al., 2025) identifies critical regions within the block-sparse pattern by applying anti-diagonal scoring to the score map. While these methods reduce computation, they predominantly rely on local Query-Key affinity scores for token selection. Critically,

these methods often overlook the causal information flow, where early tokens recursively aggregate to form deep-layer representations. Pruning them disrupts essential recursive dependencies, particularly under high sparsity ratios.

**Training-based Methods.** Recent works incorporate sparsity directly into the training or fine-tuning phase to learn optimal patterns natively. SeerAttention (Gao et al., 2024) introduces a learnable gate trained via self-distillation to select block-sparse patterns. Native Sparse Attention (Yuan et al., 2025) utilizes a differentiable selection mechanism during pre-training to optimize three distinct types of attention heads. DeepSeek-V3.2 employs Dynamic Sparse Attention (DSA), which dynamically selects top-$k$ tokens based on scores within each indexer head. Similarly, MiniCPM-4.1 adopts InfLLMv2 (Zhao et al., 2025), where top-$k$ block indices are shared across each query group. Other architectures, such as Native Hybrid Attention (Du et al., 2025) and Gated Attention (Qiu et al., 2025), improve efficiency by combining sliding windows with global blocks or by hybridizing Gated DeltaNet with standard attention through output gating. However, these approaches incur high training costs and rely on fixed sparsity patterns. Our work is **orthogonal** to these designs and can be integrated with models like DeepSeek-V3.2 or MiniCPM-4.1 to further reduce computation and accelerate the pre-filling phase without sacrificing pre-trained accuracy.

## 5. Conclusion

In this paper, we propose Stem, a unified framework designed to bridge the gap between sparse attention mechanisms and the intrinsic causal information flow of LLMs. By identifying initial tokens as critical recursive anchors, we introduce the Token Position-Decay strategy to explicitly safeguard these structural dependencies, while complementing it with the Output-Aware Metric to capture magnitude-sensitive semantic features. Whether applied as a plug-and-play module or integrated into training, Stem consistently enhances efficiency by significantly reducing prefill latency, proving that causal-aligned token selection is key to scaling the context capabilities of modern LLMs.

**Limitations and future work.** Stem currently targets prefill-stage sparse attention, where the quadratic attention bottleneck is most pronounced. For chunk prefill, its position-decay schedule can be applied by maintaining global token offsets across chunks. Extending Stem to decoding is more nuanced: OAM can naturally score tokens in the KV cache, but TPD does not directly transfer because decoding involves a single query position rather than a sequence of query positions. We leave decode-stage sparsification and long-form generation dynamics as future work.

## Acknowledgments

We gratefully acknowledge the central contributions of Lin Niu and Xin Luo to this work. Lin Niu provided close guidance throughout the project, helped shape the research direction, and contributed to the experimental design, analysis, and manuscript writing. Xin Luo led the method design, developed the main algorithmic formulation, built the experimental framework, conducted the empirical analysis, and prepared the figures. We thank Jianchen Zhu and S. Kevin Zhou for their supervision and support, and Linchuan Xie, Yifu Sun, and Guanghua Yu for helpful discussions and assistance with experiments.

## Impact Statement

This work aims to improve the efficiency of long-context language models by reducing the computational cost of the prefill stage without additional training. More efficient long-context processing may lower inference latency and resource usage, making long-document applications more accessible and reducing the environmental cost associated with large-scale model deployment. The proposed method is a general inference-time acceleration technique and does not introduce new data collection, training objectives, or model capabilities by itself. As with other efficiency improvements for large language models, it may also make existing model behaviors easier to deploy at scale, including both beneficial applications and potential misuse. We therefore emphasize that Stem should be used together with the same safety, evaluation, and deployment practices required for the underlying language models.

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

# A. appendix

## A.1. Deriving the Output-Aware Metric from a Global Objective

We derive the Output-Aware Metric (OAM) from a global reconstruction objective. For clarity, we consider a single attention head; the multi-head case follows by applying the derivation per head.

**Global objective.** Let $P \in \mathbb{R}^{N \times N}$ be the row-wise softmax attention probability matrix and $V \in \mathbb{R}^{N \times d}$ the value matrix. The dense attention output is

$$O \;=\; PV \in \mathbb{R}^{N \times d}.$$

We measure the global discrepancy to a sparse output $\tilde{O}$ using the Frobenius norm:

$$\min \; \|O - \tilde{O}\|_F.$$

**A separable surrogate bound.** To obtain an explicit selection rule, we introduce an analysis surrogate that truncates the dense output without renormalizing probabilities. For each query row $i$, let $\mathcal{A}(i)$ denote the admissible key set under masking, and choose a subset $S_i \subseteq \mathcal{A}(i)$ with a (possibly position-dependent) budget $|S_i| = k(i)$. Define

$$O_i \;=\; \sum_{j \in \mathcal{A}(i)} P_{i,j} V_j, \qquad \hat{O}_i \;=\; \sum_{j \in S_i} P_{i,j} V_j.$$

Then

$$\|O - \hat{O}\|_F = \sqrt{\sum_{i=1}^{N} \|O_i - \hat{O}_i\|_2^2} \;\leq\; \sum_{i=1}^{N} \|O_i - \hat{O}_i\|_2,$$

where we used $\sqrt{\sum_i a_i^2} \leq \sum_i a_i$ for $a_i \geq 0$. Moreover,

$$O_i - \hat{O}_i \;=\; \sum_{j \in \mathcal{A}(i) \setminus S_i} P_{i,j} V_j,$$

and by the triangle inequality (using $P_{i,j} \geq 0$),

$$\|O_i - \hat{O}_i\|_2 \;\leq\; \sum_{j \in \mathcal{A}(i) \setminus S_i} \|P_{i,j} V_j\|_2 = \sum_{j \in \mathcal{A}(i) \setminus S_i} P_{i,j} \, \|V_j\|_2.$$

Combining the above yields the global, separable upper bound

$$\|O - \hat{O}\|_F \;\leq\; \sum_{i=1}^{N} \sum_{j \in \mathcal{A}(i) \setminus S_i} P_{i,j} \, \|V_j\|_2. \tag{9}$$

Under the per-row budget $|S_i| = k(i)$, minimizing the right-hand side of (9) decouples across rows. Since all terms are nonnegative, a natural optimal choice for this bound is to retain, for each row $i$, the Top-$k(i)$ indices with the largest weights $P_{i,j}\|V_j\|_2$.

**Row-wise softmax and rank-equivalent scoring.** Let

$$s_{i,j} \;=\; \frac{Q_i K_j^\top}{\sqrt{d}}, \qquad P_{i,j} \;=\; \frac{\exp(s_{i,j})}{Z_i}, \qquad Z_i \;=\; \sum_{t \in \mathcal{A}(i)} \exp(s_{i,t}) \;>\; 0.$$

Then

$$P_{i,j}\|V_j\|_2 \;=\; \frac{1}{Z_i} \exp(s_{i,j})\|V_j\|_2.$$

For a fixed row $i$, $Z_i$ is a positive constant independent of $j$, hence it does not affect within-row Top-$k(i)$ selection. Therefore,

$$P_{i,j}\|V_j\|_2 \;\overset{\text{rank}}{\propto}\; \exp(s_{i,j})\|V_j\|_2 \;=\; \exp\!\left(\frac{Q_i K_j^\top}{\sqrt{d}}\right) \cdot \|V_j\|_2.$$

Since Top-$k$ depends only on ordering, we may apply any strictly monotone transform. Taking $\log(\cdot)$ yields the log-domain metric

$$M_{i,j} = \log\left(\exp\left(\frac{Q_i K_j^\top}{\sqrt{d}}\right) \cdot \|V_j\|_2\right) = \frac{Q_i K_j^\top}{\sqrt{d}} + \log\|V_j\|_2.$$

This motivates an output-aware scoring rule that combines routing relevance ($Q_i K_j^\top$) with value magnitude ($\|V_j\|_2$). In practice, we further apply a scaling coefficient $\beta$ (for scale matching) and a nonnegative truncation of $\log\|V_j\|_2$ to improve numerical stability, yielding the OAM used in our implementation.

