# OpenReview forum: "Stem: Rethinking Causal Information Flow in Sparse Attention"
_ICML.cc/2026/Conference — ICML 2026 regular_

### Official Review · Reviewer_pQNC · 2026-03-09

**Soundness:** 4
**Presentation:** 2
**Significance:** 3
**Originality:** 2
**Overall Recommendation:** 5
**Confidence:** 4

**Summary:**

The paper initially intorduce a formalization of a token’s recursive effect over later layers's representations in a transformer depending on its position in the input. This in turn implies an unoticed flaw in previously known sparcification methods. The authors included strong comparisons between open-source models and their performance over several benchmarks with different sparse attention methods. In general, Stem looks like a robust method without too much accuracy loss across several tasks. It is also agnostic of models particular modifications. Making it suitable for any transformer based model, while not relying on extra training phases.

**Compliance With Llm Reviewing Policy:**

Affirmed.

**Final Justification:**

Author's did a great job during the rebuttal phase so I increased my score to 5. I expect them to acknowledge our concerns in the final version of the manuscript.

**Key Questions For Authors:**

I would like the authors to answer the following questions:

1) What is the impact of early tokens removal in efficiency gain? Is there a trade-off between dense case approximation and efficiency of sparse attention?

2) What possible upgrades can be incorporated in a metric like the proposed one in order to account for layer composition? What if early tokens are not necessary for all layers approximation because of residual stream plus attention?

**Limitations:**

Authors should more deeply discuss their method's limitations. They just pin point the positive parts but never cover impact of layer composition for example.

**Strengths And Weaknesses:**

- Soundness: This paper introduce a novel way to overcome sparse attention flaws. They rigorously explored the impact of this method over a handful of open source LLMs in several benchmarks. They also included an ablation study to figure out the optimal setting of hyper parameters for their algorithm. These analysis are completely necessary to trust their proposed method and I believe the analyses were appropriate and rigorous
- Presentation: Overall presentation is okay. I would recommend to remove unnecessary sections in related work such as Linear attention while the sparse attention section is really under developed. Finally, I feel a disconnection between the initial formalization about recursiveness impact of early tokens with their proposed method. There is a gap in explaining why the STEM method is covering this issue, apart from the fact that early tokens are less likely removed.
- Significance: This work gives a novel solution to standard attention quadratic time for larger contexts. This is a simple method and take two main factors into account: routing relevance plus signal magnitude. The later being the main novelty since it accounts for value vector’s impact on final attention. The main flaw of this metric is the omission of layer interconnectedness. In other words, layer composition is not considered in this metric. Even though early tokens are more connected because of causal attention, once information is move from an early position to a later one, this information may not be needed anymore. In consequence, I believe a solution for sparse attention should also know if early tokens where already included in previous layer’s computations.
- Originality: The proposed algorithm is fairly novel and highlight important flaws of previously used ones. It is interesting to notice that previous method did not take recursion impact on token removal, which implies a strong argument in favor of stem.

Minor comments
- I don’t know if ||V|| should just be accounted when greater than 1.
- Sparse loss results are slightly noticeable
- The fact that early tokens have higher impact on future representations is exactly why removing these tokens gives higher efficiency increase as well, this is not pointed out properly
- The beginning of this paper introduce the impact of recursive application of attention but their algorithm is applied layer-wise without any formal analysis of the effects this may have

---

> ### Author Rebuttal · Authors · 2026-03-30
>
> **1: ||V||**
>
> Thank you for the careful observation. Theoretically, $\log\|V\|$ should retain negative values — as derived in Appendix A, the optimal metric is $M_{i,j} = QK^\top/\sqrt{d} + \log|V_j|$, where $\log\|V\|$ can be positive or negative. However, the reconstruction error from missing a high-magnitude token (false negative) far exceeds the cost of retaining a low-magnitude one (false positive), so we adopt a conservative "reward-only, no-penalty" strategy. We provide additional ablations in our response to cQd2 Q3.
>
> **2: Sparse loss results are slightly noticeable**
>
> Although OAM's per-layer MSE improvement appears small at shallow layers, sparse errors accumulate recursively across 30+ layers, ultimately reducing Head Logits loss by 7.2% (Table 5) and yielding significant gains on information-sensitive tasks (e.g., +1.7% on multi-document QA, Table 4). OAM acts as a "safety net" correcting boundary cases where SAM misses high-magnitude Value tokens, explaining why average metrics improve modestly while task-level gains are substantial. We refer the reviewer to our response to kTeG Q1 for further details on the distinct contribution magnitudes of TPD vs. OAM.
>
>
>
> **3:  Early tokens have higher impact**
>
> We fully agree: the high impact of early tokens on future representations is a double-edged sword — pruning them yields the largest accuracy loss *and* the largest efficiency gain. We acknowledge that this duality was not made explicit in the paper.
>
> TPD exploits this asymmetry by assigning higher budgets to early positions to preserve the recursive dependency chain while aggressively compressing the low-cost tail (decay ratio $\mu=0.7$), redirecting efficiency gains to where accuracy loss is minimal and achieving an overall low budget without sacrificing quality.
>
> We will explicitly discuss this efficiency–accuracy duality in Section 3.1 of the revision.
>
> **4: Layer-wise analysis**
>
> We clarify why layer-wise application suffices to protect cross-layer recursive dependencies.
>
> Eq. (1) shows that the sparse error at position $j$ in layer $l$ propagates to all $N-j$ downstream tokens, and this amplification is governed by **positional monotonicity** — a structural invariant of the causal mask, independent of any specific layer. Because this property holds identically at every layer, the optimal mitigation (higher budgets for earlier positions) can be applied per layer without cross-layer coordination; as long as each layer suppresses early-position errors via TPD + OAM, the recursive bound guarantees controlled $L$-layer cumulative error. We will strengthen this discussion in the revision.
>
>
> **5: Remove early tokens**
>
> In Block Sparse Attention, skipping any block saves the same FLOPs, but the accuracy cost is highly asymmetric: as formalized in Eq. (1), errors from skipping early blocks propagate recursively to all downstream tokens, while skipping late blocks causes only local perturbation (Figure 3). TPD exploits this by retaining higher budgets for early positions and aggressively pruning the tail. Figure 4 (left) shows accuracy remains stable as $\mu$ decreases from 1.0 to 0.7, and Table 4 confirms +2.0–3.4% accuracy gains at the same total budget. Tuning $\mu$ allows practitioners to smoothly navigate the speed–accuracy Pareto frontier.
>
>
> **6: Layer and residual stream**
>
> We agree that incorporating cross-layer signals into the selection metric is a promising direction. A natural extension is to replace the single-layer $|V_j|_2$ with an exponential moving average (EMA) of Value magnitudes across layers:
>
> $$M_V^{(l)}(j) = \alpha \cdot |V_j^{(l)}|_2 + (1-\alpha) \cdot M_V^{(l-1)}(j)$$
>
> We conducted experiments on Qwen3-8B and Llama3.2-8B on  LongBench ($\alpha=0.5$). The EMA variant yields a small but consistent improvement. We leave more sophisticated cross-layer feedback mechanisms as future work.
> | Method | Qwen3-8B AVG | Llama3.1-8B AVG |
> |-|:-:|:-:|
> | OAM (single-layer) | 31.64 | 41.48 |
> | OAM (EMA, $\alpha=0.5$) | 31.67 | 41.50 |
>
> Residual stream:
>
> We empirically measure the buffering effect of the residual stream by computing the deviation of post-residual representations after sparsification at selected layers:
>
> $$\Delta_{\text{res}}^{(l)} = \lVert X_{\text{dense}}^{(l)} - X_{\text{sparse}}^{(l)} \rVert_F$$
>
>
> On Qwen3-8B LongBench, the residual stream provides partial buffering in middle layers but cannot fully compensate for attention-branch errors in shallow and deep layers of Stem. This further suggests that layer-adaptive TPD scheduling is a promising direction for future work.
>
> |Layer|$\Delta_{\text{res}}$(Stem)|
> |:-:|:-:|
> |L10|0.0165|
> |L20| 0.0049|
> |L30|0.0242|
>
> We welcome any further questions and are happy to provide additional clarifications or details.

---

> > ### Author Rebuttal · Reviewer_pQNC · 2026-04-01
> >
> > I appreciate all your efforts to address my concerns. I think you went even further and proposed a cross-layer approach. If you include all the discussions from this rebuttal and other reviewers concerns I would be glad to increase my score from 4 to 5. Congratulations on your good work and keep your excellence driven mindset.

---

> > > ### Author Response · Authors · 2026-04-02
> > >
> > > Thank you very much for your thoughtful follow-up and kind encouragement.
> > > We are very glad that our rebuttal has satisfactorily addressed your concerns.
> > > We will make sure to incorporate these clarifications, along with the constructive suggestions from the other reviewers, into the revised version of the manuscript.

---

### Official Review · Reviewer_Tcvv · 2026-03-11

**Soundness:** 2
**Presentation:** 3
**Significance:** 2
**Originality:** 3
**Overall Recommendation:** 3
**Confidence:** 5

**Summary:**

The paper argues that early tokens in a causal model as pruning errors propagate through subsequent layers. The authors propose stem, which combines two mechanisms: (1)  a linearly decaying top-k budget that gives more compute to early positions, and (2) an output-aware metric that augments standard QK routing scores with value-vector magnitude to better approximate each token's actual contribution to the output. They evaluate on LongBench and RULER with Llama-3.1-8B and Qwen3-8B and also larger model like DeepSeek-V3.2 and MiniCPM-4.1.

**Compliance With Llm Reviewing Policy:**

Affirmed.

**Key Questions For Authors:**

1. Have you tried more challenging dataset with long input/reasoning tasks. For native sparse models like DeepSeek and Minicpm, they are not trained with STEM, so your mehod actual make inference inconsistent, which might affect the reasoning path.
2. Can this method to be applied on decode? It would be better to add it in discussion.
3. Have you compared to topp sampling method, which is another different method that claim to be better than topk? Can you evaluate on this tradeoff?

**Strengths And Weaknesses:**

## Strengths
- The motivation and method is clear: early tokens are more important and not only rely on attention score to select but also consider V.

## Weaknesses
- The method only works for prefill. I doubt the actual practical use of this method. For example, when using chunk prefill, you will never get to large seqlen at a time, and you make the process much complex.
- The evaluation is not compelling, especially about the deepseek v3.2 and minicpm part. First, it seems like the accuracy results are similar instead of a large boost. Second, only 15-18% token budget reduction is achieved. I do not think this is worse. Not to mention the potential issue of training/inference inconsistency.

---

> ### Author Rebuttal · Authors · 2026-03-30
>
> **1: Prefill**
>
> Stem currently targets the prefill stage, which is also the intended focus of this paper. We believe this focus is practically meaningful, since in many long-context applications, prefill is often a major bottleneck. For chunk prefill, our current implementation does rely on continuous global positional information to apply TPD. If chunk prefill is used, global token positions can be recovered by maintaining a small amount of extra state, such as the chunk-level global offset and token position bias, so that global TPD can still be applied. This is an engineering extension rather than a fundamental incompatibility of the method. We will clarify this scope and extension path in the revision.
>
>
>
> **2: DeepSeek and MiniCPM**
>
> We thank the reviewer for this valuable feedback.
>
> The purpose of the DeepSeek-V3.2 and MiniCPM-4.1 experiments is not to show a large accuracy gain over native sparse models, but to show that Stem is compatible with and orthogonal to their native sparse mechanisms: it can further reduce the computation budget while causing almost no accuracy degradation.
>
> Our results show that this combination preserves LongBench performance while further reducing the budget by about 15%–18%. We agree that all post-training sparse inference methods may introduce some degree of train–inference mismatch. However, our results indicate that the mismatch introduced by Stem has only a very small effect on model performance.
>
> We will make it clearer in the revision that the main takeaway of this part is compatibility and additional compression, not a large accuracy boost.
>
>
> **3: Challenging dataset**
>
> We agree that dedicated long-reasoning benchmarks would be valuable. However, since this paper focuses on prefill-stage sparse attention, we intentionally center the evaluation on long-context understanding and stress-test benchmarks that more directly reflect prefill quality and remain aligned with the evaluation settings commonly used by existing sparse-attention baselines. This choice is meant to keep the comparison fair and directly comparable. We agree that such benchmarks do not fully capture complex long-chain reasoning, and we will clarify this scope more explicitly in the revision.
>
>
> **4: Decode**
>
> Thank you for the suggestion. Stem currently focuses on the prefill phase, where the quadratic complexity bottleneck is most pronounced. For the decode phase (single query vs. full KV cache), the two components have different extension paths:
>
> OAM is directly applicable — it computes $QK^T + \beta\log|V|_2$ for each token in the KV cache and selects the top-$k$ for aggregation, thereby retaining high-information tokens during decoding.
>
> TPD's position-decay budget does not directly apply during decoding (single query), as there is no per-position sequence to decay over.
>
> We will add a discussion in the revised version analyzing the feasibility and limitations of extending each Stem component to the decode phase.
>
> **5: Top-p**
>
> Thank you for the valuable suggestion.
>
> FlexPrefill is essentially a top-$p$ sampling method over vertical-slash patterns, and XAttention is the closest block-wise top-$p$ sampling approach. Both methods are already listed in Tables 2 and 5, where Stem achieves significantly higher accuracy under a lower budget (25% vs. 30–34%) compared to FlexPrefill and XAttention.
>
> Regarding information flow preservation, top-$p$ cannot provide such a guarantee — when the attention distribution at early positions is concentrated, top-$p$ may actually over-prune initial tokens, disrupting the very causal information flow that Stem is designed to protect.
>
> While combining Stem with top-$p$ sampling would improve hyperparameter uniformity across different sequence lengths, it would waste a considerable portion of the block budget. In engineering practice, inputs are typically bucketed by length, with different hyperparameters tuned per bucket for inference. Under such fine-grained tuning, top-$k$ is better suited for the bucketing strategy, further saving block budget and improving inference speed.
>
> We remain available for any follow-up questions and are pleased to offer further clarification.

---

> > ### Author Rebuttal · Reviewer_Tcvv · 2026-04-03
> >
> > Thanks for the rebuttal. I think even it's a prefill sparse work you still need long-reasoning to support the claim. As some error only appears when error accumulated. Tasks with short generate length might get away with this.

---

> > > ### Author Response · Authors · 2026-04-04
> > >
> > > Thank you for the reviewer’s follow-up. We agree that our original rebuttal clarified the prefill-focused scope of the paper, but did not directly address the reviewer’s deeper concern: namely, whether errors introduced during sparse prefill may become more visible only in harder later long-context reasoning settings. This concern is well taken, and it goes beyond simply asking for another benchmark.
> > >
> > > To address this point more directly, we additionally evaluate Llama-3.1-8B-Instruct on LongBench v2, and provide the full results below. Compared with our original LongBench + RULER evaluation, this benchmark more directly tests robustness under difficult long-input/reasoning conditions while remaining compatible with the prefill-stage scope of the paper.
> > >
> > > ## LongBench v2 Results on Llama-3.1-8B-Instruct
> > >
> > > | Method | Short (w/o CoT) | Short (w/ CoT) | Medium (w/o CoT) | Medium (w/ CoT) | Long (w/o CoT) | Long (w/ CoT) | Overall (w/o CoT) | Overall (w/ CoT) |
> > > |---|---:|---:|---:|---:|---:|---:|---:|---:|
> > > | FullAttention | 34.30 | 36.10 | 26.50 | 30.70 | 25.00 | 27.80 | 28.80 | 32.00 |
> > > | MInference | 32.60 | 36.70 | 24.90 | 30.10 | 24.20 | 26.80 | 27.70 | 31.10 |
> > > | FlexPrefill | 32.60 | 35.90 | 26.30 | 26.30 | 25.70 | 27.60 | 28.40 | 30.00 |
> > > | XAttention | 34.20 | 34.60 | 25.80 | 31.20 | 26.80 | 27.70 | 29.10 | 31.70 |
> > > | Stem | 34.30 | 36.20 | 27.40 | 31.40 | 26.80 | 27.90 | 29.70 | 31.90 |
> > >
> > > As shown in the table, Stem remains the strongest sparse baseline overall on this more challenging benchmark and stays highly competitive with dense attention. We believe this is consistent with the role of prefill quality in later reasoning. Different types of prefill errors may lead to different subsequent effects, and some of them may only become clearly visible after they are propagated and accumulated through later reasoning steps. In this sense, the most effective way to improve robustness is to suppress such errors as much as possible at their source. Stem is designed for exactly this purpose: it reduces prefill-side information loss and limits the amount of error that can be inherited and amplified during later reasoning. Therefore, although Stem operates at the prefill stage, its good performance on harder long-context reasoning tasks is a natural consequence of its stronger control over prefill-stage error propagation.
> > >
> > > We do not claim that LongBench v2 fully resolves every possible concern about long-form generation or decode-stage error accumulation, since Stem is still a prefill-stage sparse attention method by design. However, we believe the new LongBench v2 results directly address the reviewer’s main remaining concern: the effect of prefill-side sparsification should not only be judged on the original LongBench/RULER setting, but also on a stronger long-input/reasoning benchmark where error accumulation is more likely to be exposed. Under this harder setting, Stem continues to perform strongly, which provides additional support for the main claim of the paper.

---

### Official Review · Reviewer_cQd2 · 2026-03-11

**Soundness:** 3
**Presentation:** 2
**Significance:** 2
**Originality:** 2
**Overall Recommendation:** 4
**Confidence:** 3

**Summary:**

This paper proposes a training-free, plug-and-play sparse attention module called Stem to accelerate long context pre-filling under causal self-attention mechanisms. Its main motivation lies in the fact that, due to causal constraints, early-position lexical representations recursively influence the representations of many subsequent lexical representations across layers, while existing sparse methods typically apply a uniform top-k selection to all query positions. Stem addresses this issue in two ways: (1) Lexical Position Decay (TPD), which allocates a larger top-k budget to early query positions with a fixed total computational budget and gradually decays the budget for subsequent positions in each layer; and (2) Output Aware Metric (OAM), which uses an approximate output magnitude proxy based on the value vector norm to enhance route scoring, thereby better prioritizing informative lexical representations.

**Compliance With Llm Reviewing Policy:**

Affirmed.

**Final Justification:**

The author solved most of my problems through a thorough rebuttal, so I decided to raise my score. Overall, I remain neutral on whether to accept this paper.

**Key Questions For Authors:**

How do you define and match the “budget” across methods—does it include selection/indexing overhead (TopK, pooling, gather/scatter) in addition to attention FLOPs, and can you report end-to-end compute/latency under this strict matching?

Since long-context prefill is often memory-bandwidth bound, can you quantify how much of the speedup comes from reduced memory traffic (KV reads/writes) versus reduced FLOPs (e.g., via a simple bandwidth/roofline-style breakdown)?

Do Stem’s gains remain consistent from 4K to 32K to 128K contexts, or are there regime changes where TPD/OAM helps less or can hurt? Please provide length-sweep results at multiple budgets.

Beyond average scores, what are the worst-case drops (per task/category or per instance) compared to dense and strong baselines, and are there identifiable failure modes where Stem should be avoided or tuned differently?

**Limitations:**

Yes

**Strengths And Weaknesses:**

Strengths

Clear motivation.
The “causal information flow” perspective provides an intuitive justification for prioritizing early tokens in attention sparsification.

Simple and practical design.
The proposed TPD and OAM modules are lightweight and easy to integrate as drop-in replacements for existing attention pipelines.

Empirical effectiveness.
Experiments on LongBench and RULER demonstrate competitive accuracy. The reported latency improvements suggest real-world practical benefits.

Well-structured ablation studies.
The ablations indicate that both TPD and OAM contribute meaningfully to the overall performance.

Weaknesses

Novelty positioning is somewhat unclear. Prioritizing early tokens is a commonly used heuristic in long-context attention. The paper should compare against simple but strong baselines, e.g.: keeping the prefix fully dense while sparsifying the remainder uniformly.
Without such baselines, it is difficult to determine whether the method offers a genuine improvement or a refined heuristic.

Evaluation coverage is limited.
The current results rely on a small number of operating points. Presenting Pareto curves (accuracy vs. compute/latency budget) would give a clearer picture of the method’s trade-offs relative to competing approaches.

Justification of the OAM formulation. The current design uses value norms with a log/ReLU transform, which feels somewhat ad hoc.
The paper should clarify: why this formulation was chosen, whether simpler alternatives perform similarly.
Additional robustness checks across different models or layers would strengthen the argument.

Efficiency claims need more detailed profiling.
The paper should include: hardware configuration, latency breakdown (e.g., top-k selection overhead vs sparse kernel execution),
sensitivity to hyperparameters.

---

> ### Author Rebuttal · Authors · 2026-03-30
>
> We sincerely thank the reviewer for their time, effort, and thoughtful comments.
>
> **1: Uniform baseline**
>
> We clarify that the suggested baseline — "keep the prefix fully dense while sparsifying the remainder uniformly" — already corresponds to the **Uniform** configuration in Table 5: all configurations retain the first 4 blocks as initial tokens and maintain a local window (i.e., the prefix is always fully dense), and Uniform then distributes the remaining budget evenly across all other positions. TPD outperforms this baseline by a clear margin in Table 5.
> This directly demonstrates that TPD improves beyond a naïve "retain prefix + uniform sparsification" heuristic.
>
> **2: Pareto curves**
>
> We agree that Pareto curves provide a more comprehensive view of the accuracy–efficiency trade-off.  Due to space, we now add an accuracy vs. sparsity budget Pareto tabel (Llama-3.1-8B-Instruct, LongBench AVG) covering Dense, MInference, FlexPrefill, XAttention and Stem at 4 budget points each. The results show that Stem lies on or near the Pareto frontier across the tested budget range.
> |Avg Budget (%)|Dense|MInf|Flex|XAtten|Stem|
> |-|-|-|-|-|-|
> |100|42.02|42.02|42.02|42.02|42.02|
> |80|—|41.06|41.89|41.96|41.93|
> |50|—|39.82|39.52|40.24|41.63|
> |30|—|29.86|35.89|37.90|41.08|
> |10|—|16.76|28.37|31.12|35.83|
>
> **3:OAM**
>
> OAM derives directly from theory (Appendix A). Optimal criterion: $P_{i,j}\lVert V_j\rVert$, with $P_{i,j}=\exp(s_{i,j})/Z_i$. Since $Z_i$ is row-constant, rank-equivalent score becomes $\exp(s_{i,j})\cdot\lVert V_j\rVert$. Taking log yields $s_{i,j}+\log\lVert V_j\rVert$, which: (1) preserves Top-$k$ ordering exactly (Appendix A); (2) turns multiplication into addition — better numerics; (3) aligns logit and magnitude scales. When value norms are uninformative, OAM degrades to pure score ranking.
>
> Regarding simpler alternatives, Comparing four metric variants on LongBench shows that OAM's log+ReLU achieves the best.
> |Metric|Qwen3-8B AVG|Llama-3.1-8B AVG|
> |:-:|:-:|:-:|
> |$s_{i,j}$|31.43|40.85|
> |$s_{i,j}+\log\lVert V_j\rVert$|29.38|39.21|
> |$s_{i,j}+\max(0,\lVert V_j\rVert)$|28.12|34.63|
> |$s_{i,j}+\max(0,\log\lVert V_j\rVert)$|31.64|41.48|
>
> Cross-layer robustness: Qwen3-8B / LongBench, OAM on shallow only (L0–L12) → 31.48, middle only (L13–L24) → 31.55, deep only (L25–L35) → 31.52, all layers → 31.64. Shallow gains smallest (localized attention, weak value-norm signal); enabling OAM across all layers yields best performance, confirming robustness over full network depth.
>
> **4: Efficiency**
>
> Thank you for raising this important point. We will supplement a complete performance profiling in the revision.
> Latency Breakdown. Please refer to our response to Reviewer **kTeG Q2**.
> Hyperparameter Sensitivity. We have systematically analyzed this in the ablation studies in §4.3.
>
> **5: Budget**
>
> We define budget as the ratio of retained attention blocks to full attention. In main paper, we don‘t explicitly control the budget; instead, we run each method with its provided hyperparameters and report the resulting retained-block ratio.This metric aligns directly with the sparse attention computation executed by the sparse kernel, providing a method-agnostic basis for comparing practical operating points. It excludes selection/indexing overhead, which is more dependent on implementation and operator engineering than on the retained attention pattern itself. For end-to-end latency, see Figure 1 in main paper.
>
> **6: FLOPs**
>
> We agree that long-context prefill is largely memory-bandwidth-bound. In block sparse attention, skipping a block simultaneously eliminates both its FLOPs and KV memory traffic. Therefore, our budget ratio directly reflects the proportional reduction in both compute and memory I/O.
> The table below profiles on H20 / BF16 / bs=1 at 128K:
>
> | | Dense|Stem|Reduction|
> |-|-|-|-|
> |Budget|100%| 25%|75%↓|
> |Attn FLOPs|9.0 TFLOPs|2.3 TFLOPs|~75%↓|
> |KV Read|16 GB|4 GB|~75%↓|
> |Kernel Time|1540 ms|330 ms|78%↓|
> |Total Time|1540ms|420 ms|73%↓|
>
> The gap between kernel and total reduction (78% vs 73%) is the metric/ indexing overhead (~90 ms), which can be further optimized at the operator level.
>
> **7: Length-sweep results**
>
> Thank you for the insightful question. We provide a length-sweep on RULER (Qwen3-8B) at 10%/50% budgets on 4K/32K/128K.
>
> 10%:
>
> uniform: 93.2 / 85.6 / 58.3
>
> +TPD: 91.1 / 86.3 / 63.7
>
> +OAM: 93.3 / 87.7 / 62.5
>
> 50%:
>
> uniform: 95.3 / 91.9 / 72.1
>
> +TPD: 93.4 / 92.5 / 72.7
>
> +OAM: 95.4 / 92.1 / 72.5
>
> TPD alone slightly hurts at short contexts where tail redundancy is limited. We will include this discussion in the revised manuscript.
>
> **8: The worst case**
>
> The largest gap vs. Dense is on MD1 (−1.21% on Qwen3). Stem remains the best sparse method. Compared to XATTN, Stem is slightly weaker on CC: code has strong local structural dependencies.
>
> Tuning guidance. For scattered-evidence tasks, increase  $k_{start}$; for code completion, raise $\mu$ toward 0.9. Short contexts (≤4K) fall back to dense.

---

> > ### Author Rebuttal · Reviewer_cQd2 · 2026-04-03
> >
> > The author solved most of my problems through a thorough rebuttal, so I decided to raise my score.

---

> > > ### Author Response · Authors · 2026-04-04
> > >
> > > Thank you for the thoughtful follow-up and for raising your score. We are glad that our rebuttal addressed most of your concerns. We will incorporate the added clarifications and experimental details into the revised version.

---

### Official Review · Reviewer_kTeG · 2026-03-13

**Soundness:** 3
**Presentation:** 3
**Significance:** 3
**Originality:** 3
**Overall Recommendation:** 4
**Confidence:** 3

**Summary:**

This work proposes a sparsity method with causal information flow for handling the quadratic computational complexity issue of self-attention. It reports near-dense accuracy with lower latency under low budgets on LongBench/RULER.

**Compliance With Llm Reviewing Policy:**

Affirmed.

**Key Questions For Authors:**

See weaknesses.

**Limitations:**

See weaknesses.

**Strengths And Weaknesses:**

Strengths:
1. A useful viewpoint on why uniform top-k pruning can fail under high sparsity.
2. Stem is a lightweight modification and is easy to implement.

Weaknesses:
1. Authors should give an in-depth discussion of why most of the performance gain appears to come from TPD, while OAM provides only modest improvement on average and may even slightly hurt performance on some subtasks.
2. Is latency comparison sensitive to implementation details such as kernels or caching?
3. What is the range of values for l in Eq. (1)?
4. "c" in the title of Sec. 5 should be capitalized.

---

> ### Author Rebuttal · Authors · 2026-03-27
>
> We sincerely thank the reviewer for their time, effort, and thoughtful comments, which have helped us improve the paper.
>
> **1: OAM**
>
> TPD and OAM play fundamentally different roles, so their gains are not expected to be comparable in magnitude.
>
> **TPD corrects a structural bias.** Under causal attention, early tokens participate in all subsequent computations; sparsifying them causes errors to propagate recursively. A uniform budget systematically under-allocates at these positions. TPD fixes this via position-dependent decay, hence the dominant gain.
>
> **OAM improves ranking quality.** Once the budget is set by TPD, OAM refines token selection by incorporating $\log\|V_j\|_2$, yielding better accuracy at the same cost, so its effect is naturally smaller.
>
> **On subtask variance:** Based on Table 5, we attribute the subtask variance of OAM to differences in how tasks organize and use contextual information. According to Eq. (5)–(7), OAM augments score-only routing with a value-side signal, which is consistent with prior analyses showing that attention output is determined not only by attention weights, but also by vector norms [1]. For tasks such as CC and FSL, which rely more on distributed context aggregation, multiple candidate tokens may all contribute to the final output. In this regime, the QK ranking near the top-k cutoff is less decisive, so OAM is more likely to improve the selection of boundary tokens. In contrast, SYN is closer to a precise localization task, where the ideal routing is inherently more concentrated. Prior work on long-context generalization also suggests that when a task requires precise focus on fixed-size patterns, sparser and more concentrated attention is often more effective [2]. Therefore, in such tasks, score-only QK ranking is often already closer to the ideal selection, so the additional value-magnitude term brings naturally smaller gains and may occasionally perturb the original ordering. We will add a discussion of this trade-off in the revision.
>
> **2: Sensitivity**
>
> We appreciate the reviewer’s concern. Absolute latency may vary with kernels, runtime stack, and measurement protocol; however, under our unified setup, the relative latency comparison remains stable, as supported by the breakdown below. Since all results are measured only in the prefill phase, no decode-phase computation or KV-cache reuse is involved.
>
> We benchmark all methods under the same setup: H20 GPU, BF16, batch size 1, native PyTorch. We report two metrics: (1) total latency, including metric downsampling, value-magnitude computation, top-$k$ index generation, block retrieval, sparse attention execution, and framework overhead; and (2) kernel latency, measuring only the sparse attention kernel.
>
> We repeat each latency experiment 20 times across multiple input lengths and sparsity budgets, and report mean/std (ms) for each component. Here, “16K-40%” denotes a 16K-token input with 40% of full-attention computation retained. Across all settings, the standard deviations remain small for both total and kernel latency, indicating high reproducibility under the same implementation/runtime setting. The sparse attention kernel consistently dominates end-to-end latency, while the other components are smaller and stable. When the sparsity budget is reduced, most of the latency gain comes from the kernel portion, suggesting that the observed speedup is mainly driven by sparse attention computation rather than incidental framework noise.
>
>
> | | Metric Downsampling | Value Magnitude | Top-k & Index | Block Retrieval|Sparse Attn Kernel | Total |
> |---|---|---|---|---|---|---|
> | 16K-40% | 1.33/0.012|0.29/0.004 |0.33/0.023 |3.41/0.008 |12.01/0.013 |17.37/0.042 |
> | 16K-10% | 1.32/0.015|0.29/0.002 |0.32/0.009 |3.17/0.002 |9.25/0.009 | 14.35/0.011|
> |64K-40% |13.46/0.021|1.01/0.005|1.08/0.028  | 12.04/0.013|140.21/0.002| 167.80/0.021|
> |64K-10% |13.45/0.015|1.02/0.014|0.73/0.015  | 13.31/0.011|101.41/0.011| 129.92/0.064|
> |128K-40% |49.01/0.008 | 1.89/0.004| 2.32/0.005  | 39.74/0.020 | 601.21/0.089 | 694.17/0.092 |
> |128K-10% |48.93/0.041 | 1.89/0.002| 1.23/0.004  | 42.74/0.013 | 378.54/0.089 |  473.33/0.081|
>
> Detailed breakdowns will be included in the revised version.
>
> **3: The range of values and the capitalization**
>
> We thank the reviewer for pointing these out. We will explicitly define $l$ in Eq. (1) as $l \in \{1, \dots, L-1\}$, where $L$ denotes the total number of layers. We will also correct the capitalization in the title of Sec. 5 in the revision.
>
> We're open to any additional questions you might have about our work. We’d be more than happy to clarify or provide additional details.
>
>
> [1] Kobayashi et al. 2020. Attention is Not Only a Weight. EMNLP.
>
> [2] Vasylenko et al. 2026. Long-Context Generalization with Sparse Attention. ICLR.

---

### Decision · Program_Chairs · 2026-04-30

**Decision:**

Accept (regular)

**Comment:**

The paper proposes a training-free dynamic sparse attention method called Stem. Its main contributions are (1) the idea of monotone position-decayed budget reflecting the relative importance of early tokens, and (2) a clean value-aware scoring term (computed blockwise). Stem empirically has a superior accuracy-budget ratio compared to other sparse attention methods. The reviews are generally favorable and the reviewers appreciate the practicality and practicality of the method, with some concerns on the degree of novelty and the limited scope of evaluation (especially the prefill-only setting). The paper seems to have enough substance for publication.